# Modifying Deutsch's scale illusion for application in music

**Issei Ichimiya** [ID] *, **Hiroko Ichimiya**

Ichimiya Clinic, Kitsuki City, Oita, Japan

* ich-oit@umin.ac.jp

## Abstract

Deutsch's scale illusion demonstrates that the overall pitch range is the preferred organization when in competition with both local (note-to-note) pitch proximity and laterality (differences in the input ear). Such intricate factors can make it difficult to mimic this illusion. If a note is under a condition in which grouping by the overall pitch range and the local pitch proximity do not conflict, we hypothesized that an illusion would be perceived simply as the result of the competition between pitch proximity and laterality. In this paper, we aimed to replicate such a condition by modifying Deutsch's scale illusion. Psychophysical studies were conducted with healthy subjects. In the first half of the study, the C major scale with successive tones was presented in ascending form, alternating between the right and left ears; counterpart notes were simultaneously presented to the opposite ear, and the subjects were asked to listen to these dichotic tone patterns. Several counterpart notes were applied; we found that when the sequences of counterpart notes were close in note-to-note pitch proximity and were not overlapped with the ascending scale in pitch, the subjects appeared to perceive the scale clearly. In the latter half of the study, we applied this condition in music and devised auditory illusions such that melodies of the passages of "Lightly Row," "Cherry Blossoms," and "Jingle Bells" were perceived by listening to "jagged" dichotic tone patterns. The method we described in this paper is simple, and it is possible to easily create auditory illusions in music by applying our method.

**Data Availability Statement:** All relevant data are within the paper and its Supporting information files.

**Funding:** The authors received no specific funding for this work.

## Introduction

Scale illusion, which was first reported by Deutsch [1–8], results from illusory conjunctions of pitch and location. The pattern that gives rise to this illusion is shown in Fig 1a and is constructed from two diatonic major scales, one ascending and the other descending, when played simultaneously. The notes presented to each ear are alternately drawn from the ascending and descending scales, giving rise to "jagged" input patterns in each ear [5]. When listening to this pattern through earphones, most subjects experience the illusion shown in Fig 1b: a melody corresponding to the higher tones is heard as coming from one earphone, while a melody corresponding to the lower tones is heard as coming from the opposite earphone. It is also interesting that listeners mostly perceive the higher stream coming from the right and the lower

**Competing interests:** The authors have declared that no competing interests exist.

**Fig 1. Schema of Deutsch's scale illusion.** (a) The stimulus is constructed from two diatonic major scales, one ascending and the other descending. (b) Through earphones, the melody patterns most subjects experience hearing while listening to (a). Few subjects perceive a full ascending or descending scale, which are shown in (c).

stream coming from the left, and the apparent locations of the higher and lower tones often remain fixed when the earphone positions are reversed.

Such effects can occur when listening to music. At the beginning of the final movement of Tchaikovsky's Sixth Symphony, the notes from the theme alternate between the first and second violin parts, and the notes from the accompaniment alternate reciprocally [6]. The

passage, however, is not perceived as it is performed; rather, one violin part appears to be playing the theme and the other the accompaniment.

Indeed, scale illusion has an intriguing effect; however, only a few variations or musical pieces creating similar illusions have been reported [6]. The intricate factors that give rise to this illusion make it difficult to mimic. Listeners, it is believed, do not perceive the "jagged" patterns that are presented to each ear because grouping by pitch proximity is powerful when in competition with laterality (i.e., differences in the input ear). Listeners instead perceive tones that are reorganized in space in accordance with their melodic reorganization; however, this explanation does not fully explain why a melody corresponding to higher tones is heard as coming from one ear, while a melody corresponding to lower tones is heard as coming from the opposite ear. If the subjects follow the pattern purely based on local (note-to-note) pitch proximity, they should hear a full ascending or descending scale [2, 4], as shown in Fig 1c, but few listeners perceive this. This observation indicates that the subjects are invoking overall pitch range as well as local pitch proximity in making grouping judgments. The strength of the overall pitch range on grouping is described in a study by Tougas and Bregman [9]. In their study, they presented a pure-tone stimulus with an "X" pattern to subjects, which consisted of two simultaneously gliding tones, one ascending and the other descending. The pattern of tones that many listeners reported was the bouncing percepts (a high glide falling and then rising and a low glide rising and then falling). Also, in a follow-up study of a Deutsch's scale illusion [5], the overall pitch range has been shown to have a large influence in grouping. In their study, the structure of the pattern of notes used in the original scale illusion study was altered slightly by adding or subtracting a pair of notes from the ends of the sequence. On hearing these notes, more listeners perceived hearing a full ascending or descending scale, but most listeners still heard the tones as two non-overlapping pitch streams.

If a note is presented under a condition in which grouping by the overall pitch range and by the local pitch proximity do not conflict, we believe that the illusion can be perceived simply as the result of the competition between pitch proximity and laterality. We have intended to demonstrate such notes in this study. In the first half of the study, the C major scale with successive tones was presented in ascending form, alternating between the right and left ears. Several counterpart notes, which were simultaneously presented to the opposite ear, were compared to find the condition in which the illusion can be perceived effectively. Further, in the latter half of the study, we applied this condition in music, and examined if auditory illusions of well-known melodies can be perceived by listening to "jagged" dichotic tone patterns. Returning to Tchaikovsky's Sixth Symphony, we will never know whether it was his intention to produce a spatial illusion, or whether he expected the audience to hear the theme waft back and forth between the two sides of the space. However, we know that the passage he created is not the only one that can produce the illusion. The method we have described in this study is simple, and it is possible to easily create more illusions in music by applying our method.

In this paper, the note pattern was drawn as a line graph, as shown in Fig 2, for easier comparison with other note patterns. Fig 2 shows the scale illusion, which is equivalent to Fig 1a and 1b.

## Experiment 1a

### Materials and methods

**Subjects.**   Twenty volunteers (11 men and nine women; mean age, 38.1 years) were included. Subjects in Experiment 1a, as well as in later experiments, had normal hearing, had no neurological conditions, and were right-handed. The study protocols in this paper were reviewed and approved by the Clinical Research Ethics Committee of Ichimiya Clinic, and the

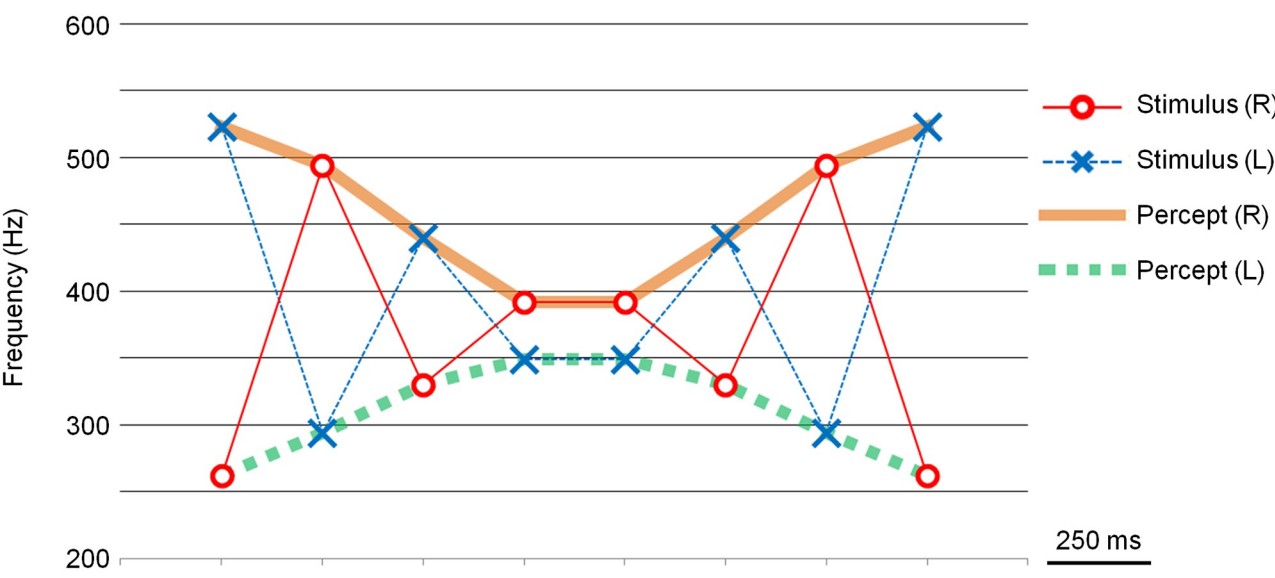

**Fig 2. Schema of Deutsch's scale illusion drawn as a line graph.** This figure is equivalent to Fig 1a and 1b and is shown here for easier comparison with other note patterns. Stimulus (R), which is shown by the red circles and solid line, represents the notes presented to the right ear. Stimulus (L), which is shown by the blue Xs and dotted line, represents the notes presented to left ear. The orange solid line and the green dotted line show the percept coming from the right and the left ears, respectively.

study was conducted in accordance with the Declaration of Helsinki. Written informed consent was obtained from all subjects prior to their inclusion in the study.

## Equipment

The same equipment was used throughout the experiments in this study. The stimulation tones were made using publicly available software, Wave Editor TWE (Yamaha Corporation, Tokyo, Japan). The tones were sinusoids of equal amplitude, and the duration of a tone component was set at 250 ms. The rise and fall times for each tone component were 10 ms, and there were no gaps between the adjacent tone components. Their frequencies ranged from 208 to 554 Hz and were set according to the 12-tone equal temperament tuning system. Tones were saved in the form of Waveform Audio Files (Microsoft Corporation, Redmond, WA, USA) with a sampling rate of 44.1 kHz/16-bit resolution.

Using an Aspire S3 computer (Acer America Corporation, San Jose, CA, USA) with a universal serial bus audio processor (SE-U55SXII; Onkyo Digital Solutions, Tokyo, Japan), auditory stimuli were delivered through dynamic headphones (MDR-7506; Sony, Tokyo, Japan) at a level of 75 dB SPL.

## Tasks

The four dichotic tone patterns in S1 Table and Fig 3 were used in this experiment. The C major scale with successive tones was presented in ascending form, alternating between the right ear and left ear. When a tone from the scale was presented to the right ear, the pitch of the tones presented to the left ear was one of the following: two whole tones lower (Fig 3a), one whole tone lower (Fig 3b), equal (Fig 3c), or one whole tone higher than that of the right ear (Fig 3d). The tonal sequence presented to the right ear was the same for all four dichotic tone patterns. When the note of the ascending scale was presented to the left ear, the tone that was two whole tones lower was presented to the right ear. The aforementioned four dichotic tone

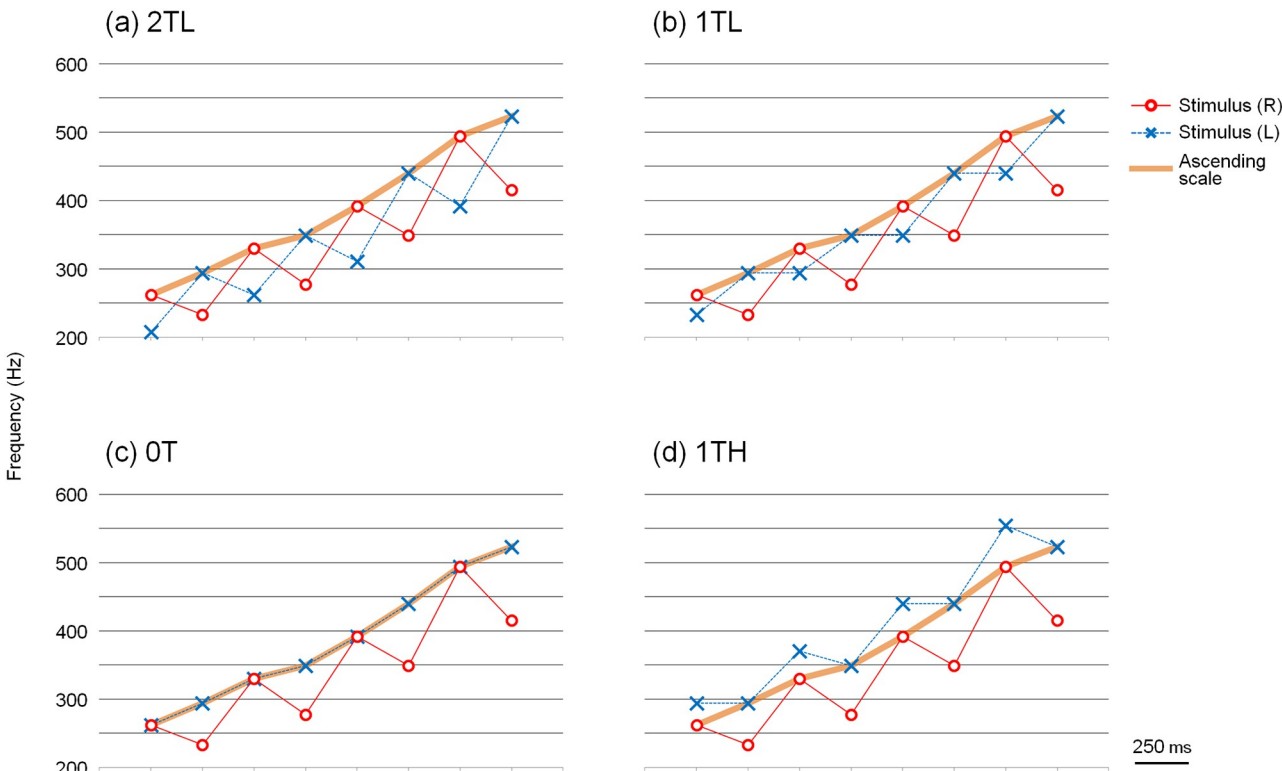

**Fig 3. Dichotic tone patterns for Experiment 1a.** An ascending scale was presented alternately to the right and left ear. Stimulus (R) and Stimulus (L) show the tone patterns presented to the right and left ear, respectively. When a tone from the scale was presented to the right ear, the pitch of the tones presented to the left ear was one of the following: (a) two whole tones lower (2TL), (b) one whole tone lower (1TL), (c) equal (0T), or (d) one whole tone higher (1TH) than that presented to the right ear.

patterns were named 2TL, 1TL, 0T, and 1TH, respectively. Theoretically, if the subjects followed the note-to-note pitch proximity, these dichotic tone patterns might have been perceived as ascending scales. If the subjects judged these tone patterns based on the overall pitch range, 2TL, 1TL, and 0T might also have been perceived as ascending scales. However, 1TH may have been perceived as a "jagged" pattern because the pitch of the left ear was always higher than that of the right ear.

Tasks were performed in a manner similar to those used in our previous study [10]. The computer monitor displayed two buttons that played two of the four above-mentioned tone patterns. The subjects were asked to click on the two buttons, listen, and choose the one where they heard the ascending scale more clearly by answering a questionnaire. They could click the buttons multiple times before responding. The tone patterns that switched between the right and left ears were tested in the same way. Thus, each subject was presented with 12 tasks. The order of the tone pairs and the order of the two buttons displayed were random.

## Statistical analysis

For each of the four tone patterns, the ratio was obtained by dividing the number the subject chose by three (i.e., the number of the tone patterns used for comparison). The values of these four ratios were compared using the Friedman test, followed by post hoc Wilcoxon signed rank test with Bonferroni corrections. A p value $< 0.05$ was considered to be statistically significant. The statistical analyses in this paper were performed using EZR version 1.52 (Saitama

Medical Center, Jichi Medical University, Saitama, Japan) [11], which is a graphical user interface for R version 4.02 (The R Foundation for Statistical Computing, Vienna, Austria). More precisely, it is a modified version of R commander designed to add statistical functions frequently used in biostatistics.

## Results and discussion

The results are shown in S2 Table. Fig 4 shows that the mean ratio of the ascending scale was perceived more clearly when compared with the other tone patterns. Because there were no statistically significant differences in the mean ratio when the right and left ears were switched, the data shown here combine the results of when the right and left ears were switched. The ratio in the case of 2TL and 0T was statistically higher than that in the case of 1TL (2TL vs. 1TL, p = 0.023; 0T vs. 1TL, p = 0.022). Interestingly, 2TL, in which the tone patterns of the ascending form and the tones that are two whole tones lower are presented alternately, is not statistically different from 0T, in which the definite ascending scale (not a "jagged" tonal sequence) is presented in one ear. Few subjects perceived the ascending scale when they heard the tonal sequence 1TH. The ratio was significantly lower than that of all the other tone patterns (2TL vs. 1TH, p = 0.002; 1TL vs. 1TH, p = 0.001; 0T vs. 1TH, p = 0.001).

In contrast to the Deutsch's scale illusion study [1, 2] that embedded both ascending and descending scales in the notes, we only embedded the scale in the ascending form. Thus, it was easy to devise dichotic tone patterns where the overall pitch range and the local pitch proximity do not conflict. From the results of this experiment, we speculated that the scale would be clearly perceived as described below. Fig 5 shows an additional line graph drawn on Fig 3. The additional line that connected the notes was the counterpart of the ascending scale, which was

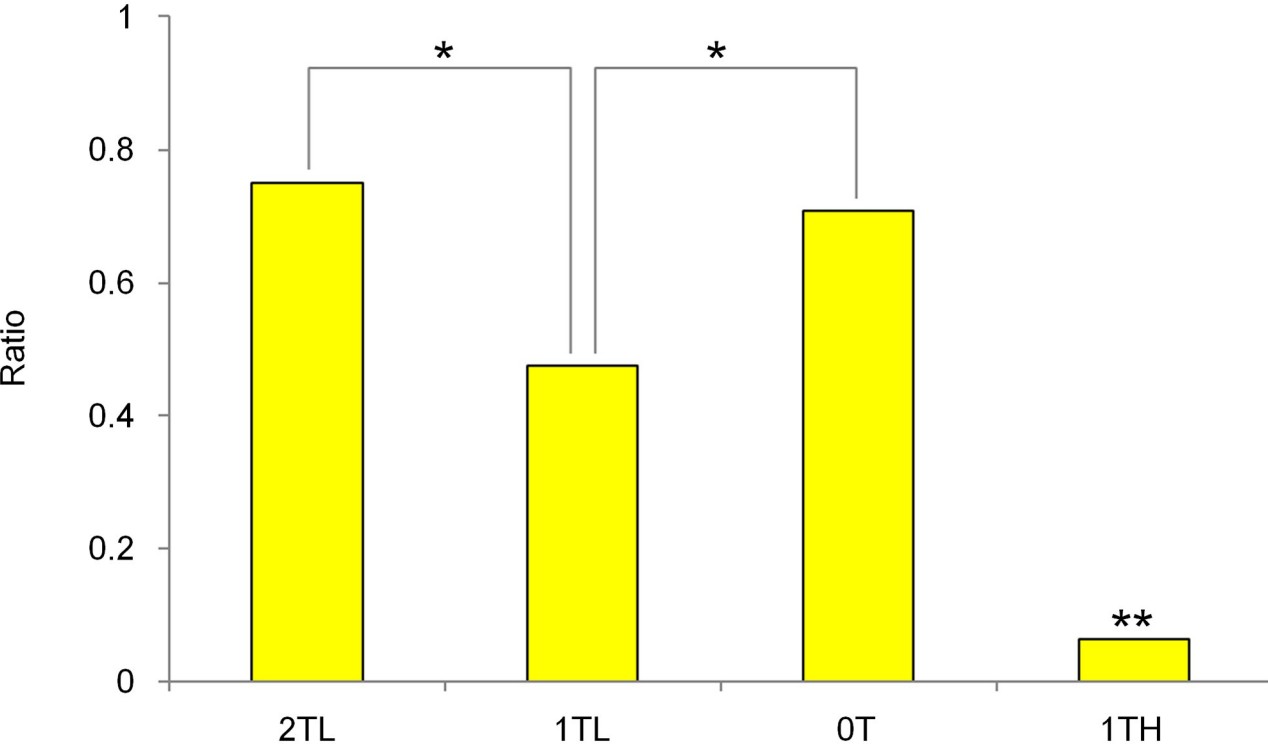

**Fig 4. Mean ratios of perception of the ascending scale in Experiment 1a.** The graph shows the mean ratio of the ascending scale that was perceived more clearly when compared with the other tone patterns. *: p < 0.05. **: p < 0.05 against all other tone patterns.

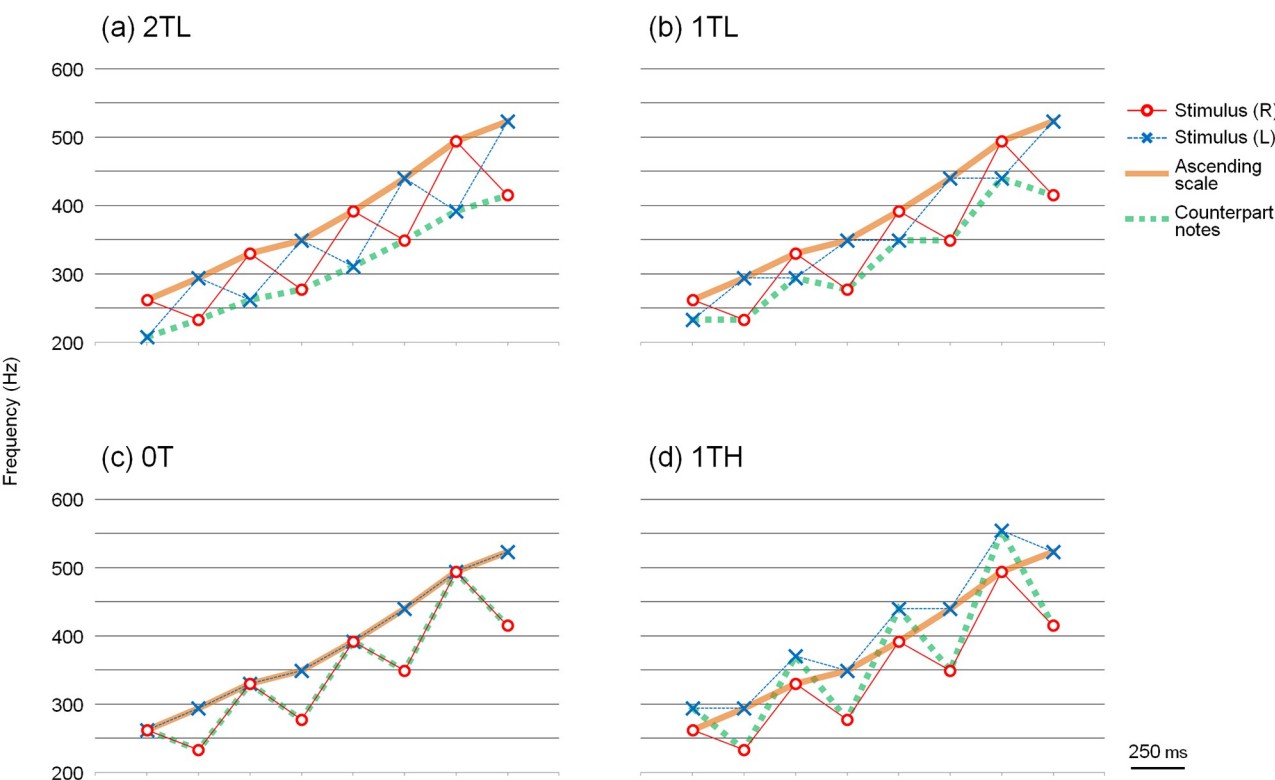

**Fig 5. Dichotic tone patterns showing the counterpart notes.** Images showing an additional line graph (green dotted line) in Fig 3. The notes that are connected with the green line are counterparts of the ascending scale that are presented alternately from opposite ears.

presented alternately from the opposite ears. The additional line of 2TL appears to be in good continuation; that is, when the sequences of counterpart notes were close in note-to-note pitch proximity, the subjects appear to perceive the ascending scale more clearly. To address this issue, another dichotic tonal sequence in which the counterpart notes would be in good continuation was examined in the next experiment.

## Experiment 1b

### Materials and methods

**Subjects.** Twenty volunteers (11 men and nine women; mean age, 36.2 years) were included.

### Tasks

The experimental protocol was identical to that of Experiment 1a, except that 1TH from Experiment 1a was replaced with the other dichotic tone patterns. The tone patterns were played simultaneously in ascending form and repeating C notes (= 262 Hz), such that whenever a tone from the ascending scale was heard in the right ear, a C note was heard in the left ear, and vice versa (S3 Table, Fig 6). These dichotic tone patterns were named CN. As can be seen from Fig 6, the ascending scale and repeating C notes do not overlap in pitch. Thus, in CN, there appeared to be no conflict between the overall pitch range and local pitch proximity.

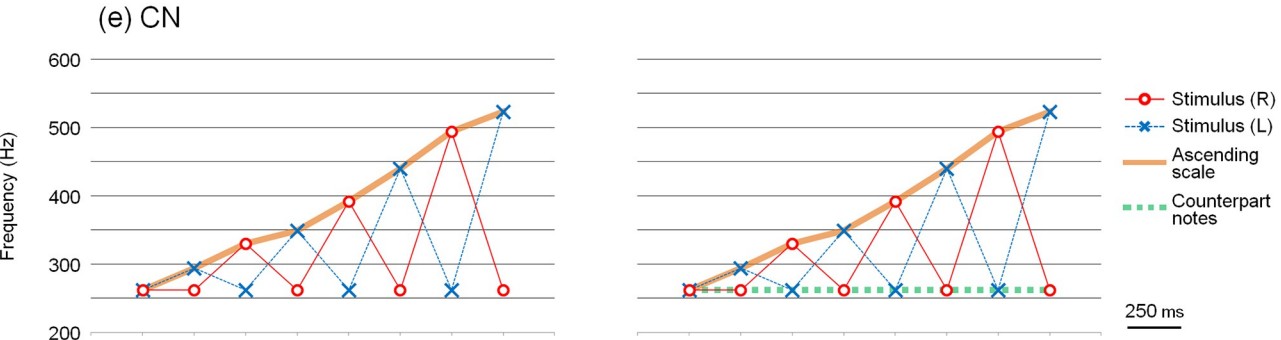

**Fig 6. Additional tone patterns for Experiment 1b.** Tonal sequence 1TH was replaced with (e), which is named CN. The additional green dotted line drawn on the right graph shows the counterparts of the ascending scale that are presented alternately to the opposite ears.

## Statistical analysis

Statistical analysis was performed in the same way as in Experiment 1a.

## Results and discussion

The results are shown in S4 Table. Fig 7 shows that the mean ratio of the ascending scale was perceived more clearly when compared with the other tone patterns. Similar to Experiment 1a, there were no statistically significant differences in the subjects' perceptions when the right

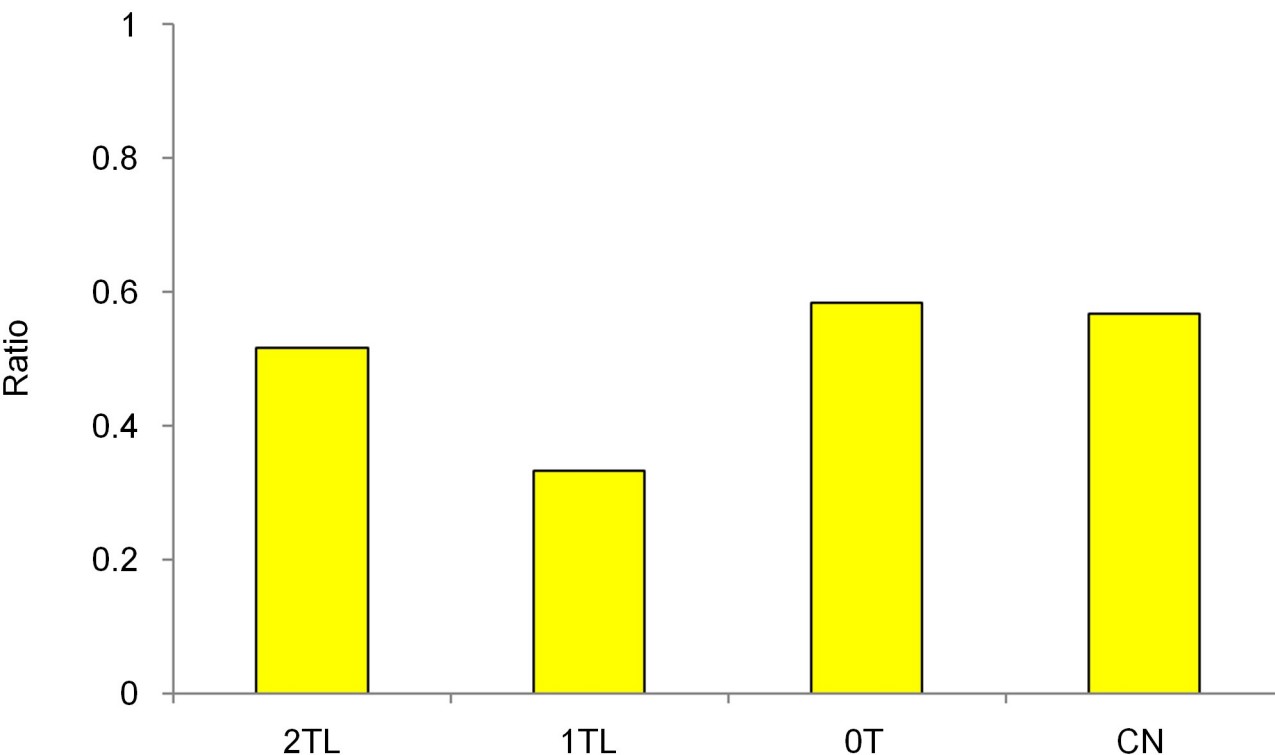

**Fig 7. Mean ratios of perception of the ascending scale in Experiment 1b.** The graph shows that the mean ratio of the ascending scale was perceived more clearly when compared with the other tone patterns. There were no statistically significant differences among the groups.

and left ears were switched. Thus, the data for when the right and left ears were switched were combined for the analysis.

There were no statistically significant differences among the groups; the ratio of CN was as high as that of 2TL. Although the ratio of 1TL appeared to be lower than that of the others, there were no statistically significant differences, as in Experiment 1a. This statistical discrepancy may have been due to the sensitivity of the analysis. The calculated ratios were not absolute numerical values. Because they were relative values that were compared with other tone patterns, the ratio was altered according to other tone patterns that were compared. Considering the combined results of Experiments 1a and 1b, it is speculated that when the counterpart notes were in good continuation, the subjects appeared to perceive the ascending scale clearly. In the following experiments, we applied the above-mentioned method to music to create auditory illusions in which passages were heard by hearing dichotic tone patterns.

## Experiment 2a

### Materials and methods

**Subjects.** In this experiment, 20 subjects (11 men and 9 women; mean age, 37.5 years) were included.

### Tasks

Passages of the songs "Lightly Row," "Cherry Blossoms," and "Jingle Bells" were used for this experiment. For each of the three passages, seven versions of tone patterns were constructed by connecting the tone components with a duration of 250 ms or 500 ms (S5 Table). Fig 8 shows the seven versions for "Lightly Row." Version (a) was a diotic version that identically played the melody for both ears (DO). Version (b) was a dichotic tonal sequence that alternately played the melody to the right ear and left ear. When the tone component of the melody was presented to one ear, a tone that was two whole tones lower was presented to the opposite ear (2TL). The versions in which only the tone patterns of the right or left channel of 2TL were heard diotically were also prepared. The versions (c) and (d) were named 2TLR and 2TLL, respectively. Versions (e), (f), and (g) were identical to (b), (c), and (d), respectively, except the counterpart tones were fixed to note C; these were named CN, CNR, and CNL, respectively. Versions of "Cherry Blossoms" and "Jingle Bells" were also prepared in the same way (Fig 9). However, because the rhythms of the passages from these three songs are different in the latter part, the passages of "Cherry Blossoms" and "Jingle Bells" were modified in rhythm to match "Lightly Row."

The computer monitor showed a button that played one of the 21 versions at random. Subjects were asked the title of the passage they heard, choosing from the following options: "Lightly Row," "Cherry Blossoms," "Jingle Bells," "Tulips," or the "ABC Song."

These passages were chosen for the experiment because they are familiar to Japanese people. "Lightly Row," which is known as the "Butterfly Song" in Japan, is a German folk song whose composer is unknown. "Cherry Blossoms" is a Japanese folk song whose composer is also unknown. "Jingle Bells" is an old American song composed by James Lord Pierpont. "Tulips" and the "ABC Song," which were not included in the passages of the present study, are also familiar songs.

### Statistical analysis

For each version, the ratio of correct answers was calculated, and these were compared with the results of version DO, which plays the genuine melody from both ears. McNemar's Chi-squared test was applied. A p value $< 0.05$ was considered statistically significant.

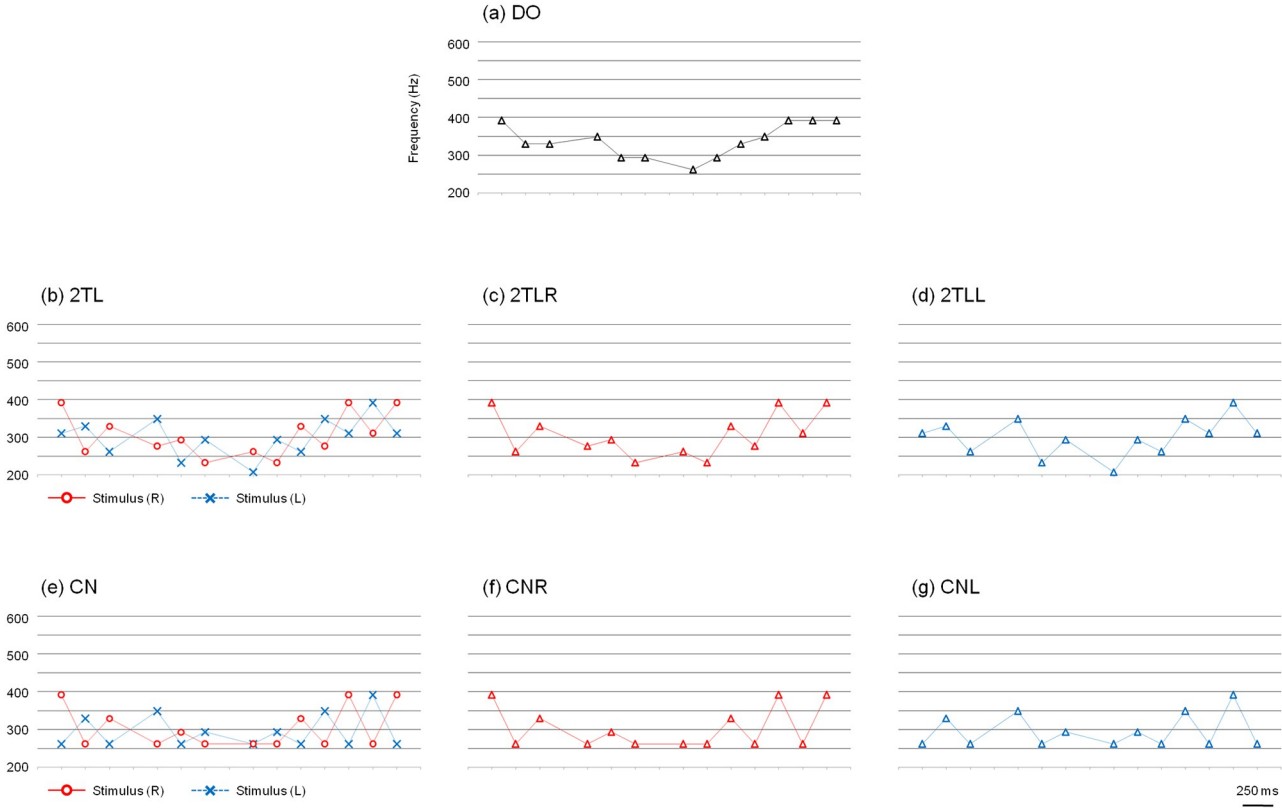

**Fig 8. Passages of "Lightly Row" used for Experiments 2a and 2b.** The graphs show 7 versions of the tone patterns constructed for these studies. (a) is a diotic version that plays the melody for both ears identically (DO). (b) is a version of the dichotic tonal sequence that plays the melody alternately to the right ear and left ear (2TL). (c) and (d) are versions in which only the tone patterns of the right or left channel of 2TL are heard diotically and were named 2TLR and 2TLL, respectively. Similarly, (e) is another dichotic version (CN), (f) is a right channel version (CNR), and (g) is a left channel version (CNL).

## Results and discussion

The results are shown in S6 Table. Fig 10 shows the mean ratios for subjects who answered the titles of the passages correctly. For all three passages, many subjects answered correctly when they heard the diotic versions (DO) of the passages. When they heard 2TL or CN, which is the version in which dichotic tone patterns were played, correct titles were also provided by many subjects. In terms of the correct answer rate, there were no statistically significant differences between DO and 2TL or CN, except for 2TL of "Jingle Bells" ($p = 0.020$). When subjects heard the "jagged" tone patterns (2TLR, 2TLL, CNR, and CNL) diotically, fewer subjects answered correctly for the titles "Cherry Blossoms" ($p = 0.001, 0.001, 0.023,$ and $P < 0.001$, respectively) and "Jingle Bells" ($p < 0.001, 0.001, 0.001,$ and $0.001$, respectively). For "Lightly Row," however, statistically significant differences were noted only for 2TLL ($p = 0.023$) and CNR ($p = 0.023$).

From these results, it may be speculated that the subjects recognized the melodies of the passages when they heard the "jagged" tone patterns dichotically. However, additional experiments that use different approaches need to be conducted to further confirm this finding, because the data in Experiment 2a were not consistent among the three passages. For "Lightly Row," some subjects were led to the correct answer when they heard the "jagged" tone patterns diotically.

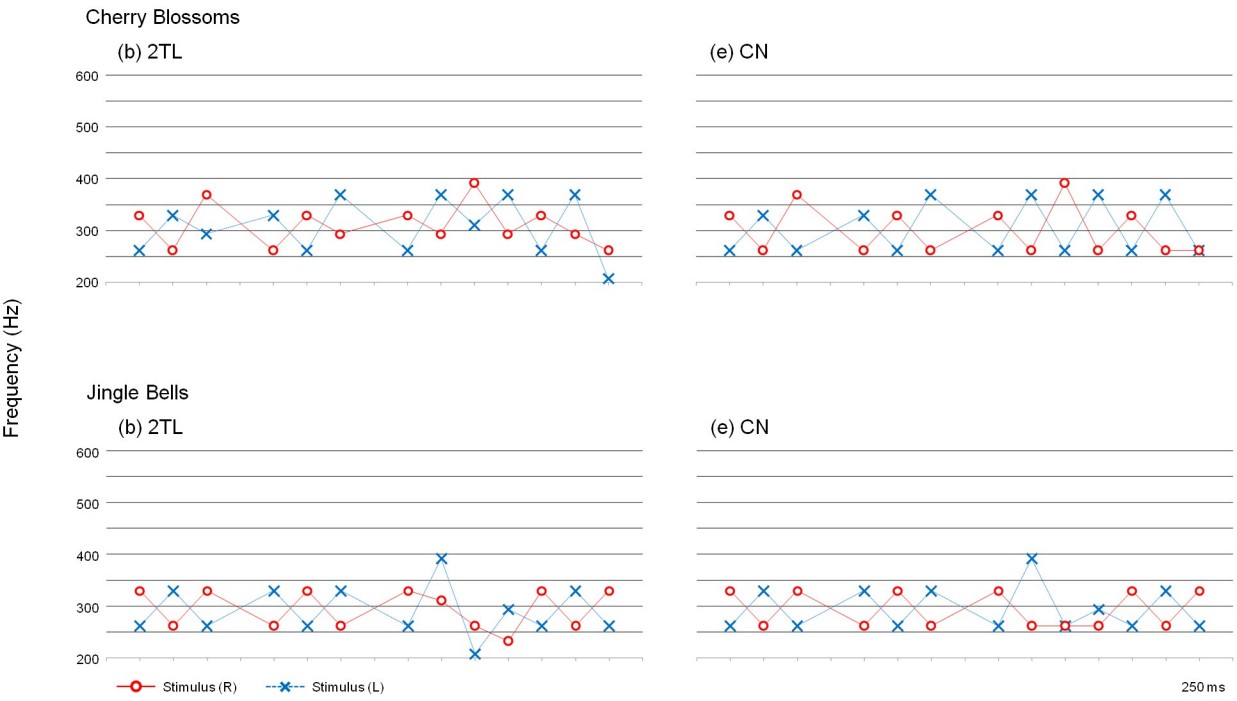

**Fig 9. Passages of "Cherry Blossoms" and "Jingle Bells" used for Experiment 2a and 2b.** (b) and (e) show the 2TL version and CN version of "Cherry Blossoms" and "Jingle Bells".

## Experiment 2b

### Materials and methods

**Subjects.** This experiment was conducted with the subjects who had previously participated in Experiment 2a. There was a time interval of at least one day between Experiment 2b and Experiment 2a. Fifteen subjects (eight men and seven women; mean age, 37.8 years) were included.

### Tasks

Using the same tone patterns as those used in Experiment 2a, tasks were performed in the same manner as in Experiment 1a. First, the tasks were performed using the versions of "Lightly Row." The computer monitor displayed two buttons that played two of the versions (DO, 2TL, 2TLR, or 2TLL). The subjects were asked to click on them and choose the melody they heard more clearly. The order of the versions and the order of the two buttons displayed was random. The tasks were also performed in the same way, using versions DO, CN, CNR, and CNL. Further, the tasks were performed in the same way using the versions of "Cherry Blossoms," as well as those of "Jingle Bells."

### Statistical analysis

For each of the four versions, the ratio was obtained by dividing the number that the subject chose by three. The values of these four ratios were compared using the Friedman test, followed by post hoc Wilcoxon signed rank test with Bonferroni corrections. A p value $< 0.05$ was considered to be statistically significant.

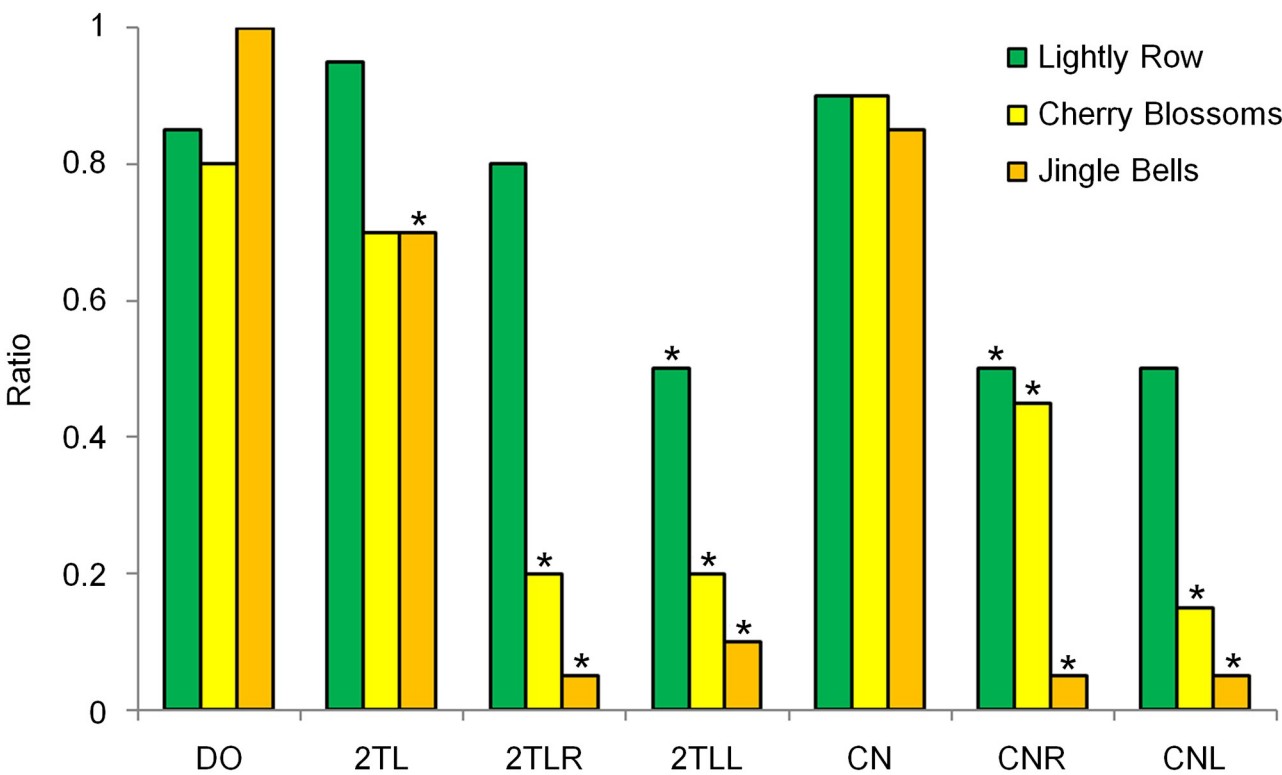

**Fig 10. Means ratios of the correct song answers.** For all three passages, many of the subjects answered correctly when they heard DO. When they heard 2TL or CN, correct titles were also provided by many subjects. When they heard 2TLR, 2TLL, CNR, and CNL, fewer subjects answered correctly for the titles "Cherry Blossoms" and "Jingle Bells," but for "Lightly Row," more subjects answered correctly when they heard these versions. Statistical analyses were performed with comparison to DO. *: p < 0.05.

## Results and discussion

The results are shown in S7 Table. Fig 11 shows the mean ratios of the melodies that were perceived more clearly when compared with the other versions. The results were almost identical, regardless of the passage. Compared to the ratios for the dichotic versions 2TL and CN, the mean ratio was statistically lower for 2TLR, 2TLL, CNR, and CNL when "jagged" tone patterns were heard diotically (for "Lightly Row," p = 0.007, 0.002, 0.003, and 0.003, respectively; for "Cherry Blossoms," p = 0.005, 0.004, 0.003, and 0.002, respectively; and for "Jingle Bells," p = 0.003, 0.003, 0.003, and 0.003, respectively). The ratio for 2TL was significantly lower than that for DO when subjects heard the "Lightly Row" (p = 0.006) or "Jingle Bells" (p = 0.027) passages. For "Cherry Blossoms," 2TL was not significantly different from DO. There were no statistically significant differences in the ratios between DO and CN for all three passages.

In the graph of Fig 11, the results of Experiment 2a were drawn as a reference. When comparing the results of both experiments, the results appeared to be similar, except for the results of "Lightly Row." In Experiment 2a, the ratio of 2TLR and CNL was so high that there was no statistically significant difference between them and the ratio for DO. In contrast, in Experiment 2b, the ratio of 2TLR and CNL was significantly lower than that of DO (p = 0.002 and 0.003, respectively).

The correct answers from Experiment 2a may not have always implied the perception of the melody. Other cues, likely several characteristic notes or the rhythms of the tone patterns, might have led to the correct answer when the subjects heard "jagged" versions. However, by

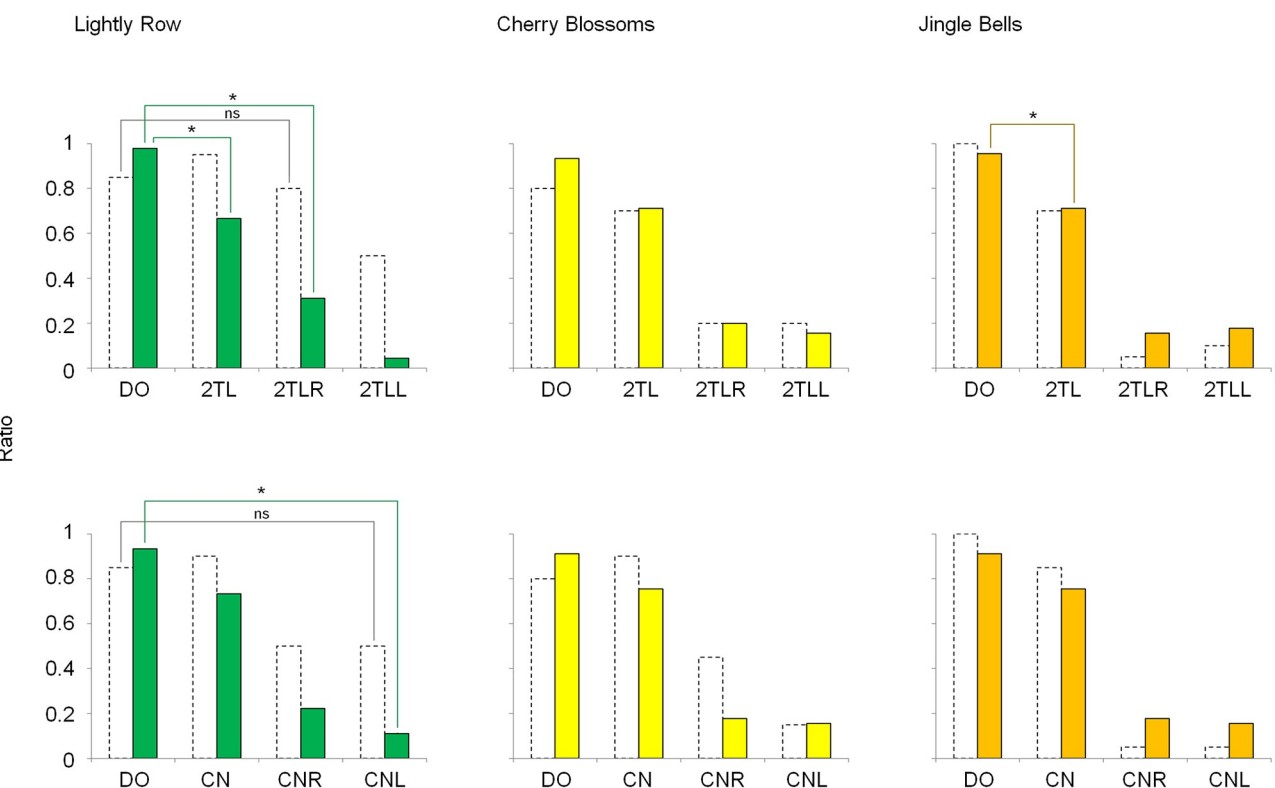

**Fig 11. Mean ratios of melodies perceived clearly when compared to other versions.** The results were almost identical regardless of the passage. Compared to ratio of the dichotic versions 2TL and CN, the ratio was lower when "jagged" tone patterns were heard diotically (2TLR, 2TLL, CNR, and CNL). The ratio of version 2TL was lower than that of DO when subjects heard the "Lightly Row" or "Jingle Bells" passage. In the case of "Cherry Blossoms," the ratio of 2TL was not statistically different from that of DO. There were no statistically significant differences between the ratios for DO and CN for all three passages. In the graphs, the results of Experiment 2a are drawn as a reference (dotted bars). *: p < 0.05. ns: not significant.

adding the results of Experiment 2b, we speculate the subjects surely recognized the melodies of the passages when they heard the dichotic tone patterns 2TL or CN. These dichotic versions were perceived as the melody, as clearly as the diotic version DO, in most cases.

## General discussion

Deutsch's scale illusion is perceived such that the overall pitch range has a larger influence in grouping compared to local (note-to-note) pitch proximity [5]. Variants of this illusion are challenging to produce since a restriction exists on the embedding of both ascending and descending scales in the notes. Deutsch [6] has also reported the chromatic and cambiata illusions, both of which are the variants produced based on the same principle as the scale illusion. The tones embedded in these illusions are chromatic scales as for the former and repeated three tones for the latter. However, we are not aware of the other reported variants with the exception of the ones that owing to the slight alteration of the structure of the pattern of notes used in the original scale [5].

Compared to the Deutsch's scale illusion, the major difference in ours is that only the ascending form of the scale was embedded in the notes. Owing to this simpler principle, the overall pitch range and the local pitch proximity could easily avoid conflict upon the production of notes. In such conditions, illusions seemed to be perceived since the note-to-note pitch proximity of the tone patterns was the preferred organization when placed against laterality.

Based on the results of Experiment 1, we devised a method to produce the notes such that the illusion could be clearly perceived, namely to produce notes wherein the sequences of counterpart notes are close in note-to-note pitch proximity, which are not overlapped with the ascending scale in pitch.

Based on the results of Experiment 2, the abovementioned method could be applied not only to a simple ascending scale but also to music. However, from a musician's standpoint, there could be room for improvement in the notes we made. Here, the counterpart tone sequences were simply set at two whole tones lower or were fixed in note C. Based on the results of the present study, however, under the condition that the musical notes of both the main melody and sub melody were in good continuation and that the main melody and sub melody did not overlap in pitch, auditory illusions were supposedly made. Upgrading the sub melody could produce illusionary notes that are more musically enjoyable. Knowledge of another illusion, which is called "octave illusion," may also be needed [12]. This illusion is produced when two tones that are spaced at an octave apart are alternated repeatedly. An identical sequence is delivered to both ears simultaneously; however, when the right ear receives a high tone, the left ear receives a low tone, and vice versa. Many people hear a single tone that switches from ear to ear, while its' pitch simultaneously shifts back and forth between high and low. Upon creating illusionary notes according to the method proposed in this paper, the main melody and sub melody should not be set an octave apart in pitch, since the effect of octave illusion has not yet been clarified in music, which is desirable to avoid this condition.

There are several auditory illusions other than scale and octave illusions. A Shepard tone [13] is a sound consisting of a superposition of sine waves separated by octaves. When played with the bass pitch of the tone moving upward or downward, it creates the auditory illusion of a tone that seems to continually ascend or descend in pitch, which ultimately becomes no higher or lower. Moreover, the phonemic restoration effect [14] is a perceptual phenomenon where, under certain conditions, sounds actually missing from a speech signal can be restored by the brain and believed to be heard. The effect occurs when missing phonemes in an auditory signal are replaced with noise that would have the physical properties to mask those phonemes, creating ambiguity. Unfortunately, these interesting illusions are not well-known. On the contrary, there are many well-known optical illusions, for example, the Rubin vase–face illusion [15], Müller-Lyer illusion [16], Zöllner illusion [17], Fraser-Wilcox illusion [18], and scintillating grid illusion [19]. Auditory illusions are more difficult to publicize than optical illusions, which can be presented in paper media. This is probably one of the reasons why auditory illusions are less popular. However, in the Internet era, it is easy to publicize sound data. We hope that by applying our method, more variants of auditory illusions will be invented and disseminated worldwide.

Finally, we would like to emphasize that the illusion in music mentioned in this paper is not only for enjoying or for musicians, because this illusion might be perceived in everyday life. Butler [3] found evidence that the perceptual reorganization that occurs in the Deutsch's scale illusion also occurs in a broad range of musical situations. Butler presented the scale illusion pattern through spatially separated loudspeakers instead of earphones and asked subjects to notate what they heard. In some conditions, the patterns were composed of piano tones, and differences in timbre were introduced between the sounds coming from the two speakers. He found that, despite these variations, virtually all responses reflected grouping by pitch proximity. This suggests that hearing pure tones from earphones is not the only condition where scale illusion is perceived. Furthermore, as our results show, hearing a simple ascending scale is not the only condition where scale illusion is perceived. The sound environment is complex, and due to reflections and reverberations, it is not easy to attribute the sounds heard to their respective sound sources. Therefore, when mixed sounds are heard simultaneously from both

ears, it is difficult to determine which component should be assigned to which source based on localization cues alone. Other factors are also at work to provide cues about different sound sources. One such factor is pitch proximity, because similar sounds often come from the same source and different sounds often come from different sources [6]; the present study has demonstrated such a possible mechanism using music. Since the effects of dichotic hearing are yet to be elucidated, further accumulating knowledge is warranted. We hope that this could eventually result in the development of new types of hearing aids or auditory rehabilitation.

## Supporting information

**S1 Table. Dichotic tone patterns for Experiment 1a.**
(XLSX)

**S2 Table. The results of Experiment 1a.**
(XLSX)

**S3 Table. Dichotic tone patterns for Experiment 1b.**
(XLSX)

**S4 Table. The results of Experiment 1b.**
(XLSX)

**S5 Table. Passages used for Experiments 2a and 2b.**
(XLSX)

**S6 Table. The results of Experiment 2a.**
(XLSX)

**S7 Table. The results of Experiment 2b.**
(XLSX)

## Acknowledgments

We thank Editage (www.editage.jp) for English language editing.

## Author Contributions

**Conceptualization:** Issei Ichimiya.

**Data curation:** Issei Ichimiya, Hiroko Ichimiya.

**Formal analysis:** Issei Ichimiya.

**Funding acquisition:** Issei Ichimiya, Hiroko Ichimiya.

**Investigation:** Issei Ichimiya, Hiroko Ichimiya.

**Methodology:** Issei Ichimiya.

**Project administration:** Issei Ichimiya.

**Resources:** Issei Ichimiya.

**Software:** Issei Ichimiya.

**Supervision:** Issei Ichimiya.

**Validation:** Issei Ichimiya.

**Visualization:** Issei Ichimiya.

**Writing – original draft:** Issei Ichimiya.

**Writing – review & editing:** Issei Ichimiya.

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
