## [Decision Letter · Decision Letter 0]

26 Apr 2022

PONE-D-21-24563Modifying Deutsch's scale illusion for application in musicPLOS ONE

Dear Dr. Ichimiya,

Thank you for submitting your manuscript to PLOS ONE. After careful consideration, we feel that it has merit but does not fully meet PLOS ONE’s publication criteria as it currently stands. Therefore, we invite you to submit a revised version of the manuscript that addresses the points raised during the review process.

The expert reviewer raises a number of concerns regarding the organization and reporting of the manuscript. In particular, they suggest significant improvements to be made to clarifications of the methodological details and justification of the rationale.

Can you please address these concerns in your revision?

We look forward to receiving your revised manuscript.

Kind regards,

Avanti Dey, PhD

Staff Editor

PLOS ONE

Journal Requirements:

Reviewers' comments:

Reviewer's Responses to Questions

**Comments to the Author**

1. Is the manuscript technically sound, and do the data support the conclusions?

Reviewer #1: Partly

2. Has the statistical analysis been performed appropriately and rigorously? 

Reviewer #1: No

3. Have the authors made all data underlying the findings in their manuscript fully available?

Reviewer #1: Yes

4. Is the manuscript presented in an intelligible fashion and written in standard English?

Reviewer #1: No

5. Review Comments to the Author

Reviewer #1: The primary challenge in reviewing this manuscript was a combination of missing information and lack of clarity. I will give overall comments as details are likely to change with any revision of the manuscript.

From the start, I'm unsure what is meant by global pitch range, local pitch proximity and laterality. I can infer the last two, but these should be explicitly defined for the context of this experiment. The Deutsch scale illusion is described but no other literature on the concepts of global pitch range, local pitch proximity and laterality, how they operate perceptually and how they interact. It is also unclear why it necessary to imitate this illusion with "music".

Throughout the manuscript, there is reference to "other notes" or "other tonal sequences". I could not determine exactly what the authors were referring to, and so it is difficult to evaluate the soundness of the reporting, or the conclusions that can be drawn. For example, the first sentence of most Results sections is meaningless without having defined the dependent variable and what the "other tonal sequences" are.

The organization of the manuscript is confusing at first. Materials and Methods are presented as if for one experiment, but later on we find that there are four. I would suggest repeating the Materials and Methods sections, along with Results and Discussion for each experiment, where the materials section for Experiments 2-4 can describe how the materials here differ from Experiment 1. The statistical analysis section is missing a description of the dependent variable and how it was calculated. I can guess at what the "mean ratio" is in the Results and Discussion sections, but it should be spelled out clearly. There is mention in the Results sections of the left and right ears being switched but this is not mentioned anywhere in the description of the stimuli or tasks.

I appreciate that the alpha was set in the Statistical Analyses section, this is important. However, when using frequentist statistics, this is the only alpha to be used, and Bonferroni correction is applied to it where relevant. This means that all p values reported should be alpha, or alpha divided by the number of tests. There is no difference in significance between .05 and .01, this is a common mistake in statistical analysis (see for example Dienes, Z. (2011) Bayesian Versus Orthodox Statistics: Which Side Are You On? Perspectives on Psychological Science 6(3)).

The description of the tasks for Experiment 4 is especially confusing. I think I understand what was done, but the description can be improved significantly.

The General Discussion is again missing any meaningful engagement with the auditory perception literature and it is still unclear why the Deutsch illusion with music is desirable or interesting. Furthermore, I would remove discussion of objective vs. subjective. First, objectivity is impossible, and the short sentence saying that subjective investigation is still meaningful comes across as being thrown in, and disingenuine. A discussion of differing frameworks and approaches however, could be interesting and meaningful here.

Specifically for the figures, I would suggest using a colour palette that is colour blind friendly - red and green are especially difficult to tell apart. The green line is sometimes missing from the figure legend.

6. PLOS authors have the option to publish the peer review history of their article (what does this mean?). If published, this will include your full peer review and any attached files.

Reviewer #1: No

---

## [Author Response · Author response to Decision Letter 0]

26 May 2022

Response to Reviewer #1:

 We appreciate your valuable comments. We hope the revised version makes it clearer to understand.

 As added in the manuscript, "local pitch proximity" means "note-to-note pitch proximity," and "laterality" means "differences in ear of input." I have changed the term, "global pitch range" to "overall pitch range" because it is less confusing. To avoid confusion, I have also changed the expression, "other notes" to "other note patterns," and "tonal sequences" to "tone patterns."

 Following your advice, I have repeated the Materials and Methods sections along with Results and Discussion for each experiment. Detailed description was added in the Statistical analysis section. In the Materials and Methods section, it is described "The tone patterns that switched between the right and left ears were tested in the same way."

 In the statistical analyses, statistical significance was set at p < 0.05, and I have deleted the description of p < 0.01. In Experiment 3, McNemar's chi-squared test was applied.

 I have changed the description of the tasks for Experiment 4.

 In the General Discussion, I have removed the discussion about objective vs. subjective because it is less important. In the Introduction and the General Discussion, I have added the description of why the Deutsch illusion with music is desirable or interesting.

 I especially appreciate your comment concerning the color of the figures because honestly, I had never thought about figures that are friendly to the color blind. For more telling apart, I have changed the green solid line to a green dotted line. I have also changed the description in the figure legend, from "red line" to "red circles and solid line," and "blue line" to "blue Xs and dotted line." The missing figure legend about the green line was also added.

---

## [Decision Letter · Decision Letter 1]

14 Sep 2022

PONE-D-21-24563R1Modifying Deutsch's scale illusion for application in musicPLOS ONE

Dear Dr. Ichimiya,

Thank you for submitting your manuscript to PLOS ONE. After careful consideration, we feel that it has merit but does not fully meet PLOS ONE’s publication criteria as it currently stands. Therefore, we invite you to submit a revised version of the manuscript that addresses the points raised during the review process.

 Thank you for revising your manuscript in light of the comments made by original reviewer. We have now had your submission reviewed by a second reviewer, whose comments are below. There is still some confusion regarding p values and Bonferroni corrections. I agree with the reviewer that clear reporting of statistical tests and results in the results section would clarify matters. Could you please carefully revise the manuscript to address all comments raised? Please submit your revised manuscript by Oct 29 2022 11:59PM. If you will need more time than this to complete your revisions, please reply to this message or contact the journal office at plosone@plos.org. Please include the following items when submitting your revised manuscript:A rebuttal letter that responds to each point raised by the academic editor and reviewer(s). You should upload this letter as a separate file labeled 'Response to Reviewers'.A marked-up copy of your manuscript that highlights changes made to the original version. You should upload this as a separate file labeled 'Revised Manuscript with Track Changes'.An unmarked version of your revised paper without tracked changes. You should upload this as a separate file labeled 'Manuscript'.If applicable, we recommend that you deposit your laboratory protocols in protocols.io to enhance the reproducibility of your results. Protocols.io assigns your protocol its own identifier (DOI) so that it can be cited independently in the future. For instructions see: https://journals.plos.org/plosone/s/submission-guidelines#loc-laboratory-protocols. Additionally, PLOS ONE offers an option for publishing peer-reviewed Lab Protocol articles, which describe protocols hosted on protocols.io. Read more information on sharing protocols at https://plos.org/protocols?utm_medium=editorial-email&utm_source=authorletters&utm_campaign=protocols.

We look forward to receiving your revised manuscript.

Kind regards,

Steve Zimmerman, PhD

Associate Editor, PLOS ONE

Journal Requirements:

Reviewers' comments:

Reviewer's Responses to Questions

**Comments to the Author**

1. If the authors have adequately addressed your comments raised in a previous round of review and you feel that this manuscript is now acceptable for publication, you may indicate that here to bypass the “Comments to the Author” section, enter your conflict of interest statement in the “Confidential to Editor” section, and submit your "Accept" recommendation.

Reviewer #2: (No Response)

2. Is the manuscript technically sound, and do the data support the conclusions?

Reviewer #2: Partly

3. Has the statistical analysis been performed appropriately and rigorously? 

Reviewer #2: Yes

4. Have the authors made all data underlying the findings in their manuscript fully available?

Reviewer #2: Yes

5. Is the manuscript presented in an intelligible fashion and written in standard English?

Reviewer #2: Yes

6. Review Comments to the Author

Reviewer #2: This is a very interesting study on the Deutsch's scale illusion. Specifically, the authors explored the condition that this illusion can be perceived effectively and they applied it in music (songs). Some of my suggestions for further improvement are as follow:

-From reading the abstract and the introduction, I thought that there are 2 experiments only. It might be clearer to replace 'in the first half of the experiment' with experiment 1 and 2 and replace 'in the latter half of the study' with experiment 3 and 4. Also, I suggest providing a brief overview of the 4 experiments in introduction.

-Line 163 and line 334-335: if p-value was set at 0.05, then the post hoc test did not use Bonferroni-corrected p value?

-It would be great if test statistics and exact p-values for significant results are reported in results session.

-Line 319-320: what is the time period between experiment 3 and 4? How soon did the participants from experiment 3 completed the next experiment?

7. PLOS authors have the option to publish the peer review history of their article (what does this mean?). If published, this will include your full peer review and any attached files.

Reviewer #2: No

---

## [Author Response · Author response to Decision Letter 1]

25 Sep 2022

Response to Reviewer Comments:

Thank you for your valuable comments. We sincerely appreciate the time and effort that you have dedicated to providing your valuable feedback on the manuscript. We have been able to incorporate changes to reflect most of the suggestions provided, and hope that the revised version is clearer to understand.

1. From reading the abstract and the introduction, I thought that there are 2 experiments only. It might be clearer to replace 'in the first half of the experiment' with experiment 1 and 2 and replace 'in the latter half of the study' with experiment 3 and 4. Also, I suggest providing a brief overview of the 4 experiments in introduction.

Response: We replaced the terms “Experiment 1, 2, 3, and 4” with “Experiment 1a, 1b, 2a, and 2b.” We have also used the expression "in the first half of the study" and "in the latter half of the study" in the revised manuscript. A brief overview of the 4 experiments was also provided in introduction.

2. Line 163 and line 334-335: if p-value was set at 0.05, then the post hoc test did not use Bonferroni-corrected p value? It would be great if test statistics and exact p-values for significant results are reported in results session.

Response: Based on the reviewer’s comments, we realized that the description regarding p values and Bonferroni corrections may have been confusing. We have clarified that the post hoc test did use Bonferroni-corrected p values, and that p values < 0.05 were considered to be statistically significant. The exact p values for significant results have now been reported in the Results sections.

3. Line 319-320: what is the time period between experiment 3 and 4? How soon did the participants from experiment 3 completed the next experiment?

Response: The time period between Experiment 2a (previously Experiment 3) and 2b (previously Experiment 4) has been specified. Experiment 2b was conducted at least one day after Experiment 2a.

---

## [Decision Letter · Decision Letter 2]

12 Dec 2022

PONE-D-21-24563R2Modifying Deutsch's scale illusion for application in musicPLOS ONE

Dear Dr. Ichimiya,

Thank you for submitting your manuscript to PLOS ONE. After careful consideration, we feel that it has merit but does not fully meet PLOS ONE’s publication criteria as it currently stands. Therefore, we invite you to submit a revised version of the manuscript that addresses the points raised during the review process.

We look forward to receiving your revised manuscript.

Kind regards,

Nicola Megna, M.D.

Academic Editor

PLOS ONE

Reviewers' comments:

Reviewer's Responses to Questions

**Comments to the Author**

1. If the authors have adequately addressed your comments raised in a previous round of review and you feel that this manuscript is now acceptable for publication, you may indicate that here to bypass the “Comments to the Author” section, enter your conflict of interest statement in the “Confidential to Editor” section, and submit your "Accept" recommendation.

Reviewer #2: (No Response)

Reviewer #3: (No Response)

2. Is the manuscript technically sound, and do the data support the conclusions?

Reviewer #2: Partly

Reviewer #3: Partly

3. Has the statistical analysis been performed appropriately and rigorously? 

Reviewer #2: Yes

Reviewer #3: Yes

4. Have the authors made all data underlying the findings in their manuscript fully available?

Reviewer #2: Yes

Reviewer #3: Yes

5. Is the manuscript presented in an intelligible fashion and written in standard English?

Reviewer #2: Yes

Reviewer #3: Yes

6. Review Comments to the Author

Reviewer #2: The manuscript is better this time, just one minor comment on reporting p-values: please report 3 decimal places for the p-values (instead of 4 decimal places) (e.g., p = .023, not p = 0.0229). Also, p = 0.000 is not possible, it may actually be 0.0000000001 and as it is too small and too long, it won't be shown on the output. So, if p is smaller than .001, you can simply report p < .001.

Reviewer #3: In my opinion, the article would benefit from a more orderly general discussion linked to the article's findings.

Typically, a discussion is expected to re-read the literature in the light of the study findings, which need to be re-described highlighting their relevance. Subsequently, authors could describe how one could proceed in that area of study.

In the discussion of this article, the authors start off confusedly with how they could improve their study (lines 386-397, for example), then list other illusions, and finally justify the choice to also use pieces of music.

I suggest an orderly and logical rewrite of the general discussion.

7. PLOS authors have the option to publish the peer review history of their article (what does this mean?). If published, this will include your full peer review and any attached files.

Reviewer #2: No

Reviewer #3: **Yes: **Nicola Megna

---

## [Author Response · Author response to Decision Letter 2]

20 Dec 2022

Response to Reviewer #1:

We greatly appreciate your valuable comments. Following your advice, we have revised the p-values.

Response to Reviewer #2:

We greatly appreciate your valuable comments. We understand that the general discussion section was confusing. Following your advice, we have re-written the section. We hope that the revised version of the manuscript has properly addressed your concerns.

---

## [Decision Letter · Decision Letter 3]

3 Jan 2023

Modifying Deutsch's scale illusion for application in music

PONE-D-21-24563R3

Dear Dr. Ichimiya,

We’re pleased to inform you that your manuscript has been judged scientifically suitable for publication and will be formally accepted for publication once it meets all outstanding technical requirements.

Kind regards,

Nicola Megna, M.D.

Academic Editor

PLOS ONE

Additional Editor Comments (optional):

Reviewers' comments:

Reviewer's Responses to Questions

**Comments to the Author**

1. If the authors have adequately addressed your comments raised in a previous round of review and you feel that this manuscript is now acceptable for publication, you may indicate that here to bypass the “Comments to the Author” section, enter your conflict of interest statement in the “Confidential to Editor” section, and submit your "Accept" recommendation.

Reviewer #2: All comments have been addressed

Reviewer #3: All comments have been addressed

2. Is the manuscript technically sound, and do the data support the conclusions?

Reviewer #2: Partly

Reviewer #3: Partly

3. Has the statistical analysis been performed appropriately and rigorously? 

Reviewer #2: Yes

Reviewer #3: Yes

4. Have the authors made all data underlying the findings in their manuscript fully available?

Reviewer #2: Yes

Reviewer #3: Yes

5. Is the manuscript presented in an intelligible fashion and written in standard English?

Reviewer #2: Yes

Reviewer #3: Yes

6. Review Comments to the Author

Reviewer #2: (No Response)

Reviewer #3: (No Response)

7. PLOS authors have the option to publish the peer review history of their article (what does this mean?). If published, this will include your full peer review and any attached files.

Reviewer #2: No

Reviewer #3: **Yes: **Nicola Megna

---

## [Editor Report · Acceptance letter]

5 Jan 2023

PONE-D-21-24563R3 

Modifying Deutsch’s scale illusion for application in music 

Dear Dr. Ichimiya:

I'm pleased to inform you that your manuscript has been deemed suitable for publication in PLOS ONE. Congratulations! Your manuscript is now with our production department. 

Kind regards, 

on behalf of

Dr. Nicola Megna 

Academic Editor

PLOS ONE